# Importance of Hydroxide Ion Conductivity Measurement for Alkaline Water Electrolysis Membranes

**DOI:** 10.3390/membranes12060556

**Published:** 2022-05-26

**Authors:** Jun Hyun Lim, Jian Hou, Jaehong Chun, Rae Duk Lee, Jaehan Yun, Jinwoo Jung, Chang Hyun Lee

**Affiliations:** 1Department of Energy Engineering, Dankook University, Cheonan 31116, Korea; iyj2368@naver.com (J.H.L.); houjimmy@naver.com (J.H.); ruri7220@naver.com (J.Y.); sfd36641@naver.com (J.J.); 2AES Tech Co., Ltd., Daejeon 34052, Korea; jaefong.chun@aestech.co.kr (J.C.); rdlee929@gmail.com (R.D.L.)

**Keywords:** hydroxide ion transport, ion conductivity, polymer electrolyte membrane, alkaline water electrolysis, hydrogen generation, impedance measurement, Zirfon, electrode configuration

## Abstract

Alkaline water electrolysis (AWE) refers to a representative water electrolysis technology that applies electricity to synthesize hydrogen gas without the production of carbon dioxide. The ideal polymer electrolyte membranes for AWE should be capable of transporting hydroxide ions (OH^−^) quickly in harsh alkaline environments at increased temperatures. However, there has not yet been any desirable impedance measurement method for estimating hydroxide ions’ conduction behavior across the membranes, since their impedance spectra are significantly affected by connection modes between electrodes and membranes in the test cells and the impedance evaluation environments. Accordingly, the measurement method suitable for obtaining precise hydroxide ion conductivity values through the membranes should be determined. For this purpose, Zirfon^®^, a state-of-the-art AWE membrane, was adopted as the standard membrane sample to perform the impedance measurement. The impedance spectra were acquired using homemade test cells with different electrode configurations in alkaline environments, and the corresponding hydroxide ion conductivity values were determined based on the electrochemical spectra. Furthermore, a modified four-probe method was found as an optimal measurement method by comparing the conductivity obtained under alkaline conditions.

## 1. Introduction

Due to the declaration of carbon neutrality, the renewable energy industry is developing significantly. The hydrogen industry is receiving a lot of attention, and the demand for hydrogen has increased significantly. Water electrolysis, which is a representative method among the methods for producing hydrogen, is classified by the electrolyte, which is a passageway for ions. Among them, alkaline water electrolysis (AWE) refers to an electrochemical process to directly convert water under alkaline conditions into hydrogen and oxygen gas through redox reactions occurring when electricity is applied [1,2]. Unlike PEM water electrolysis, AWE has the advantage that it does not require a noble metal catalyst such as platinum, and it does not use an MEA type electrolyte membrane, which is advantageous for a large area. In addition, compared to other water electrolysis industries, technology commercialization has progressed a lot. Here, AWE is run under alkaline conditions, which are formed by adding a certain amount of strong bases (e.g., KOH or NaOH) to water [3,4]. It can activate the dissociation of hydroxide ions from aqueous strong bases with alkalinity and contribute to electro-catalytic reactions at energy levels close to the theoretical voltage (i.e., 1.23 V) by lowering overpotentials, particularly ohmic overpotential that is generated during AWE reactions. Thus, the polymer electrolyte membrane is one of the vital components of AWE, which is capable of preventing direct contact with the anode and cathode, providing the hydroxide ion conductivity between the electrodes, and separating the generated gases [5,6,7,8,9,10,11]. In particular, the membrane requires high hydroxide ion conductivity to consume a low voltage per current density and efficient hydrogen production in AWE [12,13,14,15,16,17,18,19]. Over the past few years, numerous research groups have been committed to increasing the hydroxide ion conductivity, which has been found as an important electrochemical performance relating to the whole AWE process [20]. However, the existing analyses have been conducted only at the cell terminal of the AWE unit cells, and there has been rare research on the hydroxide-ion-conductive membrane [10]. Thus, the specific electrochemical analysis of the hydroxide-ion-conductive membrane should be urgently conducted under the actual AWE operating conditions [9].

For this purpose, several measurement methods have been developed and used to assess the electrochemical performance of the membrane. However, a measurement of the hydroxide ion conductivity of the membrane has not yet been well established, and it is generally measured indirectly by immersing the membrane in DI water [9,21]. The above measurement methods are different from the actual AWE operation conditions, so it is difficult to reflect in practice. In addition, the AWE is commonly operated under high temperatures (above 60 °C) and strong alkali conditions (30 wt.% KOH) [8,22]. The stainless material applied in the AWE cell incurs corrosion strongly under the above harsh operation conditions, and all corrosion causes resistance, thus resulting in unreliable electrochemical assessment results. For direct comparison, chemical resistant material AWE cell and a method to measure hydroxide ion conductivity under the same AWE operation conditions should be developed and applied [23,24]. Thus far, the two-probe (H-type cell) method and the four-probe method for measuring hydroxide ion conductivity have generally been used [25,26]. The two-probe method is simple and convenient and has mainly been used to measure the total impedance of electrochemical cells containing anodes, cathodes, membranes, electrolytes, and organic or inorganic fillers, etc. [27]. Accordingly, the impedance of this two-probe method will reflect all the impedance components in the path of current flow (e.g., lead inductance, lead resistance, as well as capacitance between two leads) [28]. As a result, there is a limitation since typical impedance measurements should only be applied to sample materials exhibiting high impedance of 10 k ohm or more, and only other impedance components with relatively low values compared with 10 k ohm can be ignored. In contrast, the four-probe method is capable of reducing the effects of load impedance (e.g., lead inductance, lead resistance, and capacitance), since it uses four separate leads, and no current exists in the leads for sensing voltage. For measurement accuracy, it can be used in the high impedance measurement range, as well as in the low impedance range of lower than 1 ohm [27,28].

Inspired by the above analysis, this study purposed to collect more accurate information for the measurement of the hydroxide ion conductivity of the anion exchange membrane (AEM). A precision impedance measurement system including an electrode system was developed to measure and compare the resistance of membrane samples based on the two-probe method and the four-probe method in a temperature- and humidity-controlled chamber with the use of a commercial AWE membrane (e.g., Zirfon^®^). In particular, the resistance was measured in an innovative four-probe cell built of polyether ether ketone (PEEK) materials and nickel-coated stainless use steel (SUS) material with chemical resistance to harsh alkaline solution and high temperature stability under actual AWE operation conditions. Furthermore, the hydroxide ion conductivity of Zirfon^®^ was determined and compared in different KOH concentrations (1–8 M) and ambient temperatures (30–90 °C) using the modified four-probe cell method.

## 2. Materials and Methods

### 2.1. Materials

Zirfon^®^ (Agfa, Mortsel, Belgium), which is typically used in AWE, is a porous composite membrane material composed of a polysulfone matrix and ZrO_2_ present as a powder and was adopted as a reference membrane [12]. It was pretreated in boiling deionized water for 1 h to remove certain residual organic molecules, then immersed in a specific concentration KOH solution for 24 h before measurement. Potassium hydroxide (KOH, purity > 93%) was purchased from Daejung Chemical (Siheung, Korea). All the chemicals in the reagent grade were used as received without any further purification.

### 2.2. Measurements

The electrode system was employed with three types of cells (H-type cell, zero gap cell for High-Frequency Resistance, and 4-probe cell) to determine the impedance of each Zirfon^®^ membrane sample. All the membrane samples were cut to fit the size of the respective electrochemical cell and immersed in 30 wt.% KOH solution for one day before the impedance measurement.

#### 2.2.1. Hydroxide Ion Conductivity of Specific Concentration KOH Solution

The hydroxide ion conductivity of a 1–8 M concentration KOH solution at 30–90 °C was measured for comparison based on a conductivity meter (SevenExcellenceTM, Mettler Toledo, Columbus, OH, USA) and a conductivity sensor (Cond probe InLab 731-ISM, Mettler Toledo, Columbus, OH, USA).

#### 2.2.2. The 2-Probe Cell Method (H-Type Cell and Zero Gap Cell)

The impedance of the Zirfon^®^ membrane in the H-type cell (VB8-S, EC-FRONTIER, Kyoto, Japan) was measured using a 2-probe method that included platinum used as a working probe and a counter probe, respectively. A circular Zirfon^®^ membrane with a radius of 1 cm was placed between the two probes, which avoided direct contact between the two probes, and a 30 wt.% KOH solution at the respective probe around the membrane was arranged, as presented in Figure 1. The distance between the electrodes (E_d_) was set to 8 cm. Subsequently, the impedance was measured in situ at frequencies from 1 Hz to 20 kHz with the use of a potentiostat (VMP3, Bio-logic, Seyssinet-Pariset, France). Lastly, the hydroxide ion conductivity value was determined by Equation (1):Ion conductivity, σ (S/cm) = l/(R × S)(1)
where R denotes ohmic resistance; l represents the distance between electrodes; S expresses the active area of the membrane.

A zero gap cell (CNL Energy Co., Gimpo, Seoul, Korea) was adopted, and Figure 2 depicts the cell design. The distance between the electrodes (E_d_) was relatively short compared to the H-type cell and was set close to zero. The impedance of Zirfon^®^ membrane was also measured using the above 2-probe method. First, the blank resistance value was determined by measuring the resistance of the unit cell without a membrane with a single channel potentiostat under 30 wt.% KOH aqueous solutions at 30–90 °C. Subsequently, the membrane was applied to the unit cell and the total resistance was measured in the same way at the same concentrations of KOH solutions. Lastly, the result of subtracting the blank resistance value from the total resistance value was adopted to determine the hydroxide ion conductivity by Equation (1).

#### 2.2.3. The 4-Probe Cell Method

A shielding system was applied with alkali-resistant PEEK and nickel-coated SUS in the unit cell to prevent changes in the KOH concentration due to temperature and humidity and to maintain the membrane’s equilibrium state (by immersing the cell in the KOH solution or not), as presented in Figure 3 and Figure 4. Before the impedance measurement was performed, the membrane was fixed with six knurled nuts and a Teflon cover at a constant pressure to maintain the same contact resistance. The measurement was performed after the respective hydrated Zirfon^®^ membrane in the impedance measurement system reached a controlled humidity and temperature.

The impedance of the respective Zirfon^®^ membrane was measured using an AC-impedance 4-probe method in the frequency range of 100 kHz to 100 MHz at a constant current of 0.01 mA. The new 4-probe cell was manufactured with a PEEK material, which was highly stable in a KOH solution, and the probe component was installed with a platinum wire. The experiment was performed after the installation in a shielded heat- and humidity-controlled chamber for stable measurement. The impedance of a membrane sample at the controlled humidity and temperature was measurable based on a Nyquist plot [29,30]. The resistance of the sample was calculated according to the value at which the imaginary component of impedance converges to zero in the high-frequency region, and it was used to obtain the conductivity by Equation (1).

## 3. Results and Discussion

### 3.1. Aqueous KOH Solution Hydroxide Ion Conductivity

Figure 5 presents the measured hydroxide ion conductivity of aqueous KOH solutions as a function of temperatures and concentrations using the conductivity meter (SevenExcellenceTM, Mettler Toledo, Columbus, OH, USA). With the increase in the temperature and the concentration, the hydroxide ion conductivity of aqueous KOH solutions tended to increase. The reason is that the amount of ions would increase with the increase in the concentration, and the mobility of ions would increase with the increase in the temperature, thus leading to an increase in ionic conductivity [31]. Comparing the increase rate of the ion conductivity, the increasing rate with the concentration was confirmed more significantly than the increase rate with the temperature. As revealed by the above results, the hydroxide ion conductivity of aqueous KOH solutions could be more dependent on concentration than temperature.

### 3.2. The Two-Probe Cell Method

Figure 6 depicts a Nyquist plot of the impedance measured using an H-type cell. The impedance consisted of a real part and an imaginary part, and it can be expressed as Equation (2). Equation (2) suggests that the impedance of the imaginary part was affected by the frequency. In the high-frequency region, the Z′ value at which the value of Z″ converged to 0 is interpreted as the membrane resistance (R).
(2)$$\mathrm{Impedance}\;\left(Z\right)=\;Z\prime \;-\;\mathrm{jZ}\prime \prime \;=\frac{R}{1+\omega^{2}C^{2}R^{2}}-j\frac{\omega CR^{2}}{1+\omega^{2}C^{2}R^{2}}$$
where *ω* denotes angular frequency, *C* is capacitance in the parallel circuit, and *R* is resistance.

Figure 7 depicts a Nyquist plot of the impedance measured using a zero gap cell. Figure 7a is a graph showing the Nyquist plot of the impedance measured at 30 °C among column (2) in Table 1. Column (2) of Table 1 is the resistance value measured when the membrane was fastened in the zero gap cell, and it is expressed as the total resistance including the resistance of the electrode and lead wire. The resistance of the membrane was determined using the difference between the above two resistances, as listed in column (4). Column (4) is applied to Equation (1) to calculate ionic conductivity (column (7)), and area resistivity (column (8)) is calculated by Equation (3).
Areal specific resistance, ASR (Ω cm^2^) = t/σ(3)
where σ denotes ionic conductivity; t represents the thickness of the membrane.

Figure 8 depicts the measured hydroxide ion conductivity of Zirfon^®^ in 30 wt.% KOH solutions in the H-type cell and zero gap cell using the two-probe method. The hydroxide ion conductivity value of Zirfon^®^ obtained by the H-type cell was 0.08 S/cm at 30 °C, significantly lower than the hydroxide ion conductivity value of the aqueous KOH solution at 30 °C (0.66 S/cm). The above difference in values was due to the resistance of different elements between the anode and cathode in the H-type cell. It is generally known that the membrane resistance of Zirfon^®^ is approximately 0.3 Ω cm^2^ at 30 wt.% KOH solution and 30 °C conditions [12,32]. If this value is converted into a conductivity value by Equation (3), a value of 0.16 S/cm will be obtained.

Nevertheless, the result measured in the H-type cell was 0.08 S/cm, significantly lower than the commonly known value of 0.16 S/cm at 30 wt.% KOH solution and 30 °C. This result suggested that not only the effect of the resistance value of Zirfon^®^, but also all resistances, including probes and KOH solutions, were recognized and measured in the above method. During the measurement of impedance using the two-probe method in the H-type cell, the impedance measured included the impedance of all organic–inorganic properties between the anode and cathode. As a result, it was confirmed that the hydroxide ion conductivity was lower than that of the known value.

Moreover, the hydroxide ion conductivity result of Zirfon^®^ measured in a zero gap cell suggested that a value of 0.1 S/cm was calculated at 30 °C. This value was closer to the theoretical value (0.16 S/cm) than the value measured in the H-type cell. The difference between the two types of cells was based on the space between the electrode and the membrane, which would increase the impedance. The above increased the error in the electrochemical impedance spectroscopy (EIS) measurement [28]. Since the electrode-to-electrode distance in the zero gap cell is shorter than that of the H-type cell, the impedance value has a generally higher accuracy than the H-type cell [26]. However, the above method is also a two-probe-based measurement method, and it is still difficult to measure actual hydroxide ion conductivity due to the effect of different sources of electrochemical cell resistance, including resistance to the membrane as well as interfacial resistance between the membrane and the electrode and the load.

### 3.3. The Four-Probe Cell Method

Figure 9 shows a Nyquist plot of hydroxide ion conductivity measured using the four-probe method. In mode 3 (Figure 3), an abnormally low resistance was measured by immersing the membrane in the KOH solution. From the perspective of electrochemistry, the ions moved in a pathway with relatively low resistance, so they migrated through the KOH solution, instead of through the membrane. Additionally, the resistance decreased significantly (e.g., parallel connection of resistors). As depicted in Figure 3, the KOH solution was used as a parallel resistance in the measurement using the four-probe method by placing the membrane in the KOH solution. Subsequently, the hydroxide ions first moved through the KOH solution with relatively small resistance instead of the membrane. Under the above condition, the resistance decreased as the resistances were connected in parallel, as presented in Figure 3. Thus, it is difficult to conclude that the measured value is the resistance value of the membrane.

In addition, a certain level of resistance was measured in the mode, as illustrated in Figure 4 in which the membrane was immersed in KOH solution. The resistance factor of the membrane was well reflected, since the movement of hydroxide ions moved through the membrane, instead of through the KOH solution.

Figure 10a depicts the hydroxide ion conductivity measured using the four-probe method by immersing the test cell in the KOH solution. The resistance of the membrane was significantly lower and resulted in super-high hydroxide ion conductivity values, probably due to the hydroxide ions moving first and rapidly through the KOH solution rather than the membrane. However, as depicted in Figure 4, the modified cell was installed with a wetting membrane immersed in a specific KOH solution before the measurement and measured in the same four-probe method. Unlike the measurement as illustrated in Figure 3, there was a connection of a linear series resistor, and its total resistance is defined as Equation (4) [28]:R_Total_ = R_a_ + R_m_ + R_a_(4)

The measured impedance was employed for Equation (1) to obtain the ion conductivity of the sample. In other words, resistance was measured and recorded as impedance (R_m_) at the Y-intercept on the real axis of the impedance curve within the high frequency [27,28]. Here, R_a_ is a value that can be collected and ignored as “0” in the high-frequency region, so R_Total_ = R_m_ can be established. Accordingly, the areal resistance of the Zirfon^®^ membrane was measured at 0.31 Ω cm^2^, consistent with the well-known value in the specification of Zirfon^®^ [12,15,32]. Furthermore, the calculation of hydroxide ion conductivity using the determined resistance was confirmed as 0.16 S/cm, as presented in Figure 10b. In contrast to other methods, since the current path and the voltage sensing cable are independent of each other in the four-probe method, the induced impedance can be ignored from the total impedance.

Figure 11 shows the resulting graph of the Zirfon^®^ membrane impedance measurements performed using the modified four-probe cell (Figure 4 mode), and the wetting membranes were immersed in 1–8 M KOH solution before the measurement under the same thermal and humidity control conditions, such as 30–90 °C and 70% relative humidity (RH). As the temperature and concentration increased, the hydroxide ion conductivity value increased, similar to the ion conductivity variation results of KOH solutions as illustrated in Figure 5.

In brief, the proposed four-probe method (Figure 4 mode) of the methodologies analyzed above was found as the most appropriate method in accuracy and precision. The two-probe method cannot measure the intrinsic impedance of the membrane, since it contains all the forms of resistance including the MEA, bipolar plate, and endplate, etc. Thus, it is not effective in measuring the specific resistance of the membrane. In addition, the modified four-probe method can measure only the intrinsic impedance of the membrane using an independent electrode circuit [28]. Thus, the above modified four-probe method can lead to more reliable measurement results of the impedance of an AEM.

## 4. Conclusions

In this study, different electrochemical test methods were used, and the results were compared and investigated to find the optimal measurement method of the membrane’s actual hydroxide ion conductivity. First, after the conductivity was measured as a function of the concentration and temperature of the KOH solution, the transport behavior of hydroxide ions was confirmed to be more dependent on the effect arising from concentration than on the effect of temperature. Moreover, the hydroxide ion conductivity was calculated using two types of electrochemical methods (including the two-probe method and the four-probe method). In all the measurements, Zirfon^®^ membrane was a reference sample, and the impedance was measured after each Zirfon^®^ membrane immersed in KOH solution was stabilized to reach the equilibrium moisture content. The impedance measured using the two-probe method had a rather high resistance than the known resistance value, since all elements in the path from the test sample to the electrode through the potentiometer were included as impedance components. The above conclusion was confirmed, as the resistance value measured in the zero gap cell with a relatively short distance between the electrodes was smaller than the impedance measured in the H-type cell. In addition, the four-probe method can measure the intrinsic impedance of a membrane using an independent electrode circuit. However, during the measurement of the resistance of the membrane in the KOH solution (Figure 3), the hydroxide ions did not move through the membrane but moved through the KOH solution with relatively low resistance, so it could not be effective in measuring the resistance of the membrane. During the measurement (Figure 4), hydroxide ions moved through the membrane, so the correct resistance value to be measured could be calculated. The resistance value measured using the above method was consistent with the known resistance value of the commercial membrane. Thus, the electrochemical cell using the modified four-probe method was found as the most suitable method to measure an AEM’s hydroxide ion conductivity.

## Figures and Tables

**Figure 1 membranes-12-00556-f001:**
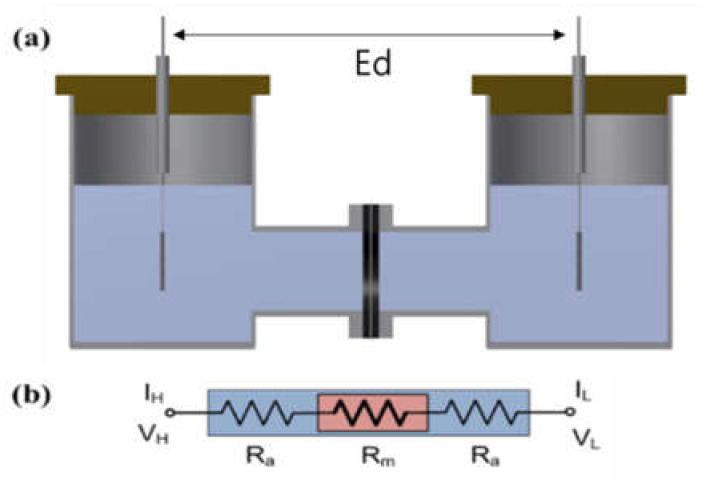
(**a**) H-type cell and (**b**) electric circuit: membrane immersed in a KOH condition. (V_H_: voltage high; V_L_: voltage low; I_H_: current high; IL: current low; R_a_: ohmic resistance of an alkaline solution; R_m_: ohmic resistance of a membrane coupon; Ed: electrode to electrode distance.)

**Figure 2 membranes-12-00556-f002:**
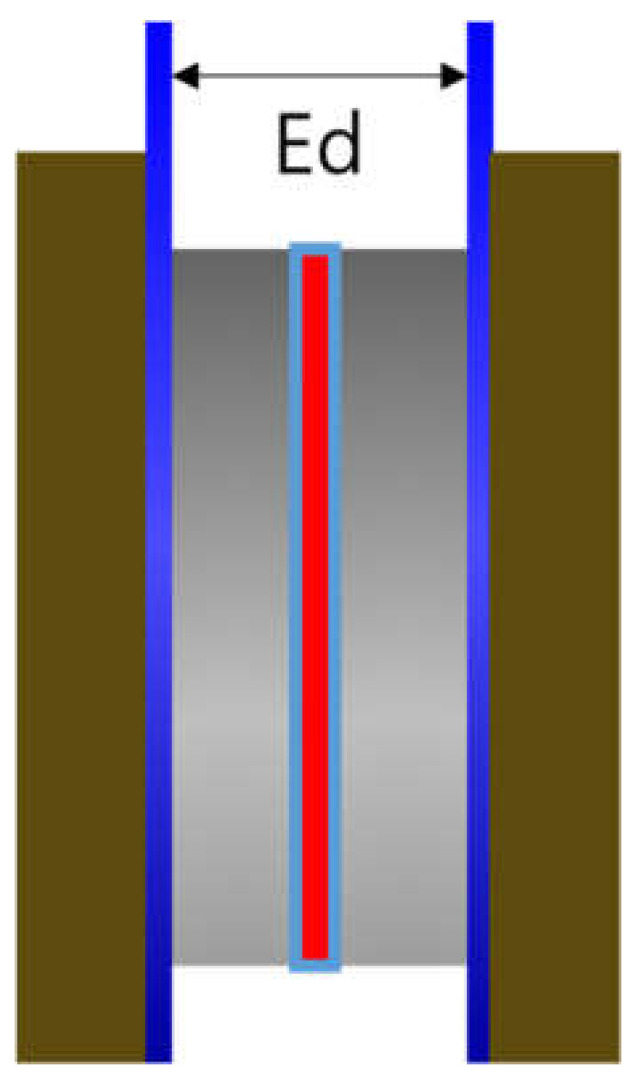
Zero gap cell employing a membrane immersed in a KOH condition (Ed: electrode to electrode distance).

**Figure 3 membranes-12-00556-f003:**
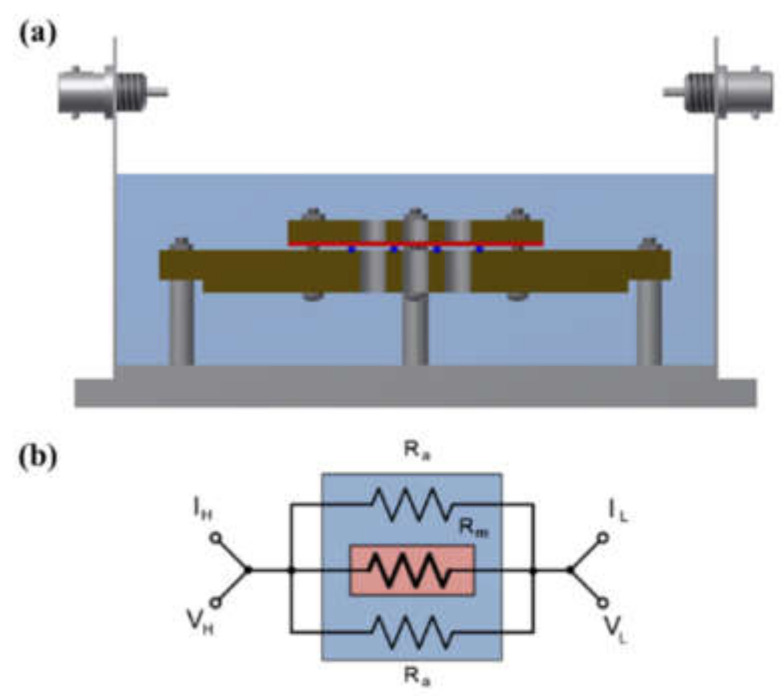
(**a**) Four-probe cell and (**b**) electric circuit: the cell immersed in a KOH solution. (V_H_: voltage high; V_L_: voltage low; I_H_: current high; IL: current low; R_a_: ohmic resistance of an alkaline solution; R_m_: ohmic resistance of a membrane coupon; Ed: electrode to electrode distance.)

**Figure 4 membranes-12-00556-f004:**
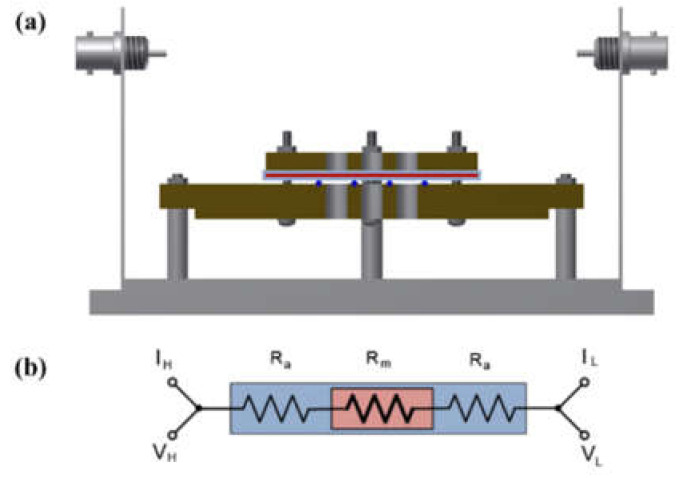
(**a**) Modified four-probe cell and (**b**) electric circuit: the cell without immersing in a KOH solution only installed a wetting membrane. (V_H_: voltage high; V_L_: voltage low; I_H_: current high; IL: current low; R_a_: ohmic resistance of an alkaline solution; R_m_: ohmic resistance of a membrane coupon; Ed: electrode to electrode distance.)

**Figure 5 membranes-12-00556-f005:**
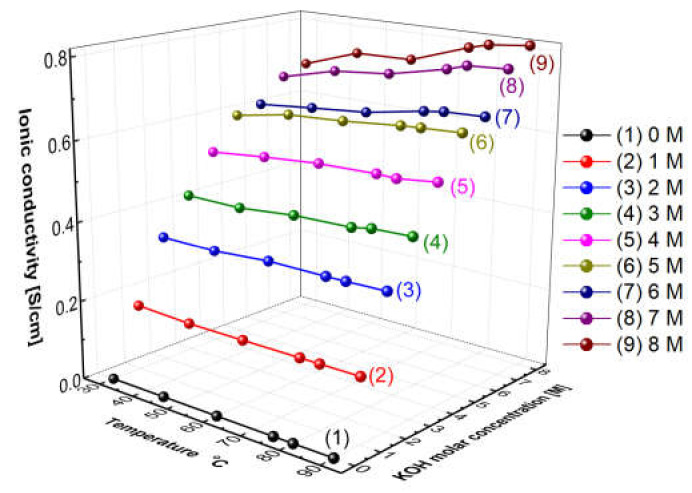
Hydroxide ion conductivity of aqueous KOH solutions as a function of temperature and concentrations.

**Figure 6 membranes-12-00556-f006:**
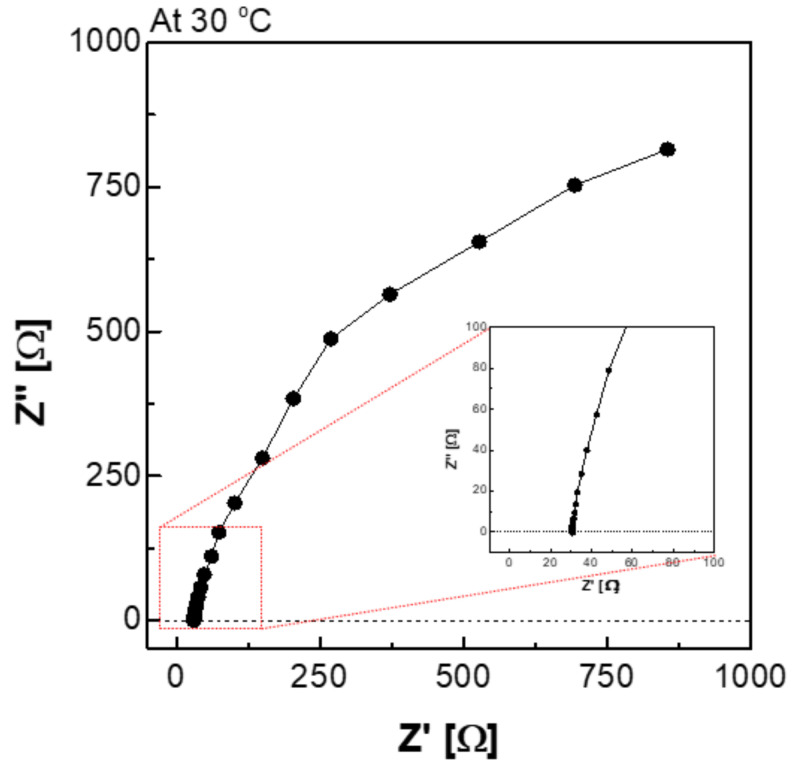
Nyquist plot via the H-type cell in 30 wt.% KOH solution at 30 °C condition.

**Figure 7 membranes-12-00556-f007:**
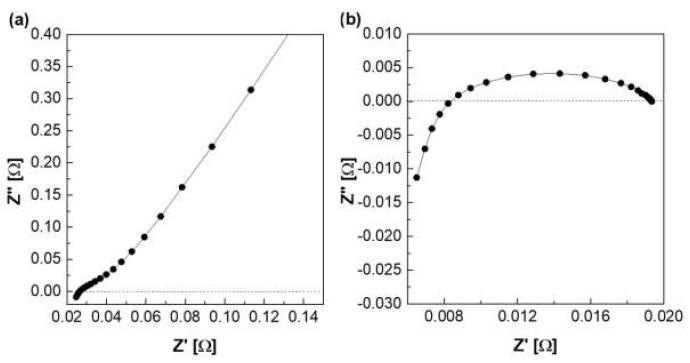
Nyquist plot via the zero gap cell in 30 wt.% KOH solution at 30 °C (**a**) with membrane, (**b**) without membrane.

**Figure 8 membranes-12-00556-f008:**
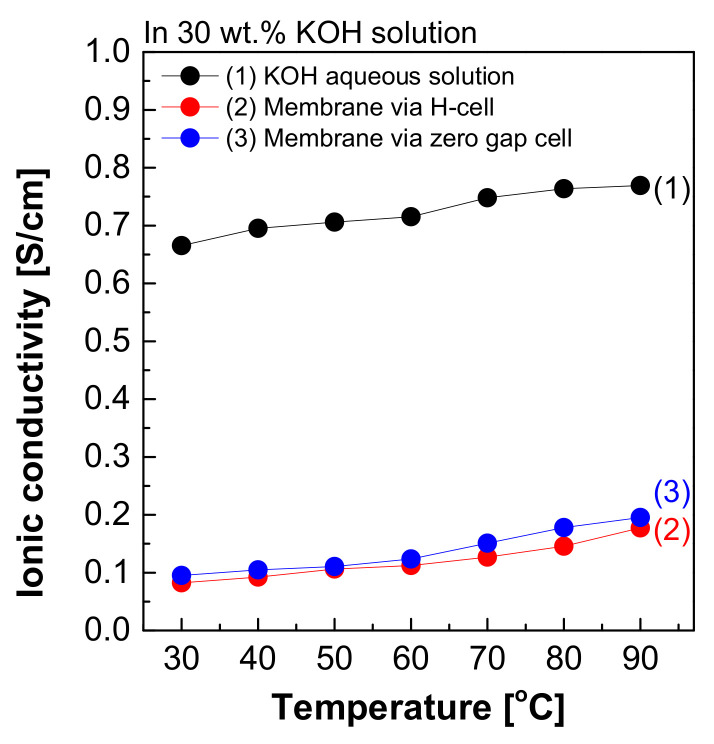
Hydroxide ion conductivity measured using the 2-probe method in 30 wt.% KOH solution.

**Figure 9 membranes-12-00556-f009:**
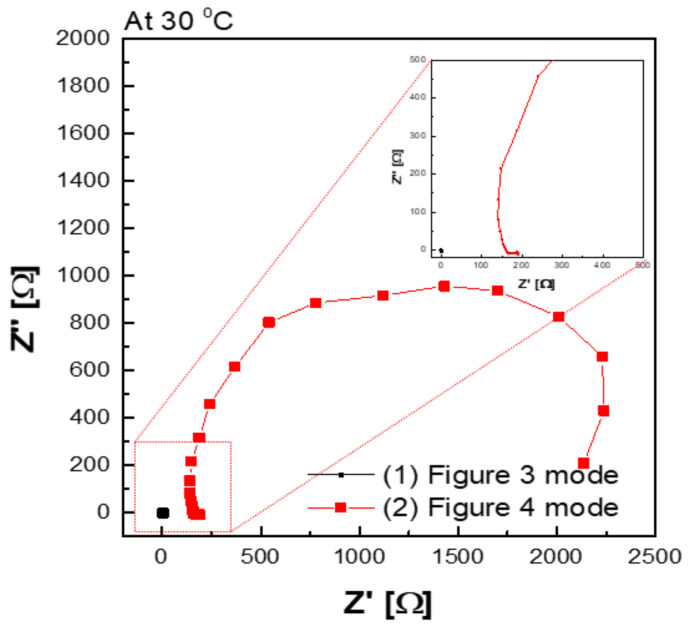
Hydroxide ion conductivity measured using the 4-probe method in 30 wt.% KOH solution immersing under 30 wt.% KOH condition.

**Figure 10 membranes-12-00556-f010:**
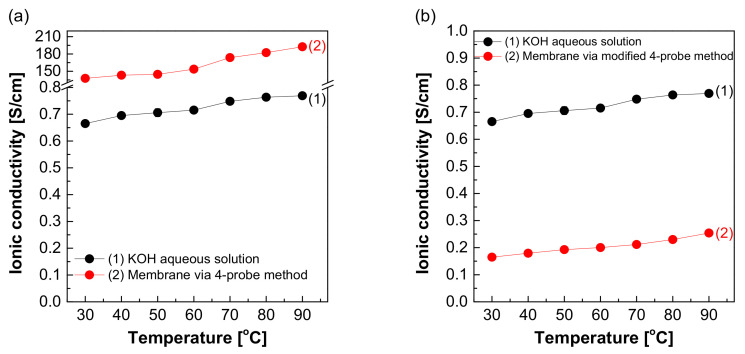
Hydroxide ion conductivity measured using the 4-probe method in 30 wt.% KOH solution (**a**) immersing in 30 wt.% KOH condition (Figure 3 mode), (**b**) without immersing in 30 wt.% KOH solution (Figure 4 mode).

**Figure 11 membranes-12-00556-f011:**
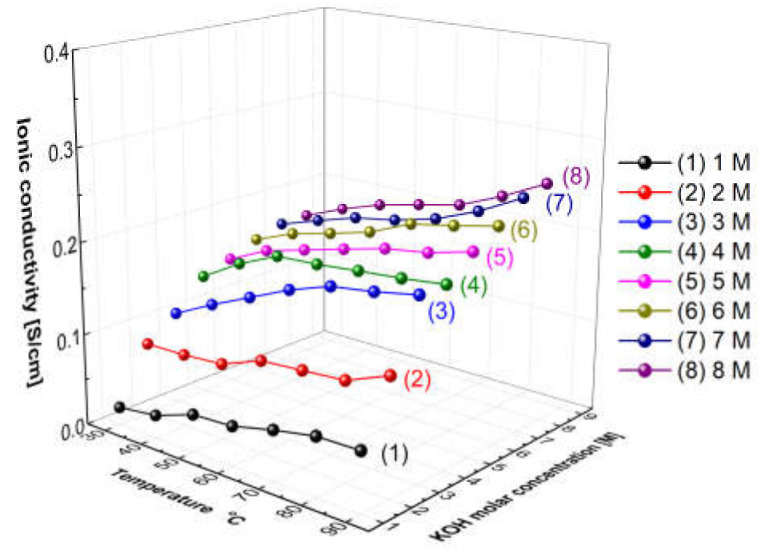
Hydroxide ion conductivity measured using the 4-probe method (Figure 4 mode) without immersing in a KOH condition only installed wetting membranes immersed in 1–8 M KOH before the measurement.

**Table 1 membranes-12-00556-t001:** Electrochemical properties via zero gap cell.

Zirfon (in 30 wt.% KOH)							
(1) Temperature (°C)	30	40	50	60	70	80	90
(2) Total resistance (Ω)	0.023	0.021	0.019	0.018	0.016	0.013	0.011
(3) Blank resistance (Ω)	0.0081	0.008	0.008	0.0079	0.0078	0.0078	0.0075
(4) Membrane resistance (Ω) ((2)−(3))	0.015	0.013	0.011	0.010	0.008	0.005	0.003
(5) Thickness (μm)	498						
(6) Active area (cm^2^)	25						
(7) Ion conductivity (S/cm)	0.0953	0.1048	0.1107	0.1237	0.1509	0.1779	0.1953
(8) Area resistance (Ω cm^2^)	0.523	0.475	0.450	0.403	0.330	0.280	0.255

## Data Availability

The data presented in this study are available on request from the corresponding author.

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
