# Peer review of "Importance of Hydroxide Ion Conductivity Measurement for Alkaline Water Electrolysis Membranes"

_membranes, 2022, doi:10.3390/membranes12060556_

Round 1

Reviewer 1 Report

The manuscript “Importance of Hydroxide Ion Conductivity Measurement for Alkaline Water Electrolysis Membranes” by Lee and coworkers presents good alkaline conductivity data for the alkaline electrolysis of water. The manuscript needs to be revised thoroughly, especially in the structure, experimental data and discussion before publication in Membranes.  

  • Page 3: the subsection 2-probe cell needs more expatiation about experimental measurements and the material used for the working and counter probe.
  • Page 4: is there the subsection 2.2.3 missing from the manuscript? check your sections and subsections numbering system
  • Page 4: Stability study of the used material in the experimental conditions is very important. Is there any stability study for the 2-prob and 4-prob cells and the used membrane at different KOH concentrations and temperatures?
  • Figure 5: once can see that the effect of temperature is only observed above 6o C and concentration of 4 M. is this due to the lack in stability of the membrane and cells at these conditions??At 3M it seems that conductivity decrease when temperature increases from 80 t0 90 C, any explanation for that?
  • Page 5: subsection number 3.1 has been repeated two times?? please check your sections and subsections numbering system.
  • Page 6: for Figures 6 and 7, why temperature 30 is chosen for measurements?
  • Page 7, during the discussion of Figure 8, it is mentioned that the "conductivity value of aqueous KOH solution at 30 oC (0.66 S/cm)." as shown on the figure. However, the comparison later with the value of 0.16 S/cm. which makes some confusion for the readers. Make your comparison in discussion the figures clearer and concise.
  • Page 9: Figure 10: in Figure 10 b the condition on the figure shows that the measurements are "in 30 wt% ..." while in the caption it sys " without immersing in 30 wt%...." please check your experimental data and conditions in all of your experimental and discussion all around the manuscript
  • References need to be updated with 2022 relevant literatures

Author Response

Please, check the attached file

Reviewer 2 Report

The manuscript reported the suitable impedance measurement method for estimating hydroxide ions’ conduction behaviour across the membranes. In this work, Zirfon® was adopted as the standard sample to perform the impedance measurement with different electrode configurations in alkaline environments. Furthermore, the conductivity data were compared with those acquired under membrane-free alkaline conditions to find the optimal measurement method.

I suggest giving a minor revision and the authors need to clarify some issues or supply some more experimental data to enrich the content. This could be a comprehensive and meaningful work after revision.

See the attached PDF file for detailed comments.

Author Response

please, check the attached file

Round 2

Reviewer 1 Report

The manuscript has been substantially improved and can be published in Membranes.